# The Use of Artificial Intelligence in Orthopedics: Applications and Limitations of Machine Learning in Diagnosis and Prediction

Bernardo Innocenti *[ID], Yanislav Radyul and Edoardo Bori [ID]

BEAMS Department, Ecole Polytechnique de Bruxelles, Université Libre de Bruxelles, 1050 Brussels, Belgium
* Correspondence: bernardo.innocenti@ulb.be

**Abstract:** Over the last several years, the impact of Artificial Intelligence on the world and on society has been undeniable. More specifically, a subfield, known as Machine Learning (ML), is driving innovation in a vast variety of fields as it denotes the ability of a machine to identify relationships between data without explicit criteria, emulating a human-like type of learning. Over the last decade, research efforts have also been focused on orthopedics in order to provide help and assistance to surgeons and clinicians in their daily tasks. The purpose of this paper is to serve as a guide by presenting the most recent research and achievements in orthopedics concerning these new technologies, by exposing the main concepts and limitations of different applications, and tackling the main problems concerning both the field and the technology itself. The main ML techniques will be introduced and qualitatively explored, by considering the indexes that better identify the performance of the models; then, the main two applications will be addressed: diagnosis and prediction. Finally, a discussion about the limitations of the studies and technologies will be proposed.

**Keywords:** artificial intelligence; IA; machine learning; diagnosis; orthopedics

## 1. Introduction

Over the last several years, the impact of Artificial Intelligence (AI) on the world and on society has been undeniable. More specifically, a subfield of AI, known as Machine Learning (ML), has been driving innovation in a vast variety of fields. ML is the most tangible manifestation of AI [1] and denotes the ability of a machine to identify relationships between data without explicit criteria [2], emulating a human-like type of learning. Over the last decade, research efforts have been focused on a large variety of topics, and orthopedics represents no exception, since ML has the potential to revolutionize the field of orthopedic surgery [3]. The purpose of this paper is to serve as a guide by presenting the most recent studies and achievements in orthopedics concerning AI and ML, and by exposing the main concepts and limitations of such type of technology by tackling the main problems concerning both the field and the technology itself.

## 2. Machine Learning: Concepts and Techniques

All the techniques that will be addressed in this paper (XGBoost, CNN, DCNN and Efficient-Net) are based on supervised learning [4], in which a model containing many free parameters is trained using previously labeled data (training data set) to form associations [3].

There are two main problems tackled by supervised machine learning: *classification* and *regression*. Classification is a type of problem where the goal is to identify which class the input belongs to amongst a predefined list of possibilities [5,6]; as an example, a model could be finalized to detect the presence of a bone fracture basing itself on a radiography (binary classification). The output is therefore a discrete one, and can assume only one of a limited amount of predefined values. The opposite is instead true for regression: this type of task indeed has the goal of predicting a continuous number [5,7]; an example of such

tasks could be a model that has to predict the length of stay of a surgery patient based on their age, height and weight.

Once the type of problem is defined, training has to be performed by minimizing a loss function. The loss function has as input the error between the actual output of the model and the desired output, and thus measures how close the first is to the latter. The learning process consists in adjusting the gains values (commonly called "weights") on the nodes [8] between layers by using methods related to loss function's gradient, in order to minimize the error and thus the loss function.

An important and frequently found concept in learning and training [8–11] is that of *hyper-parameters*. These are attributes that can be related to network topology and regularization issues, such as, for example, the previously mentioned model's loss function [9], or the number of feature maps in a convolution layer (see the relative paragraph for further info). Usually, different combinations of these parameters are tested to find the best ones in terms of performance, which can be quantified by different indexes (see next paragraph).

The Convolutional Neural Network (CNN) is a type of neural network where the convolution operation is one of the fundamental building blocks [12]. The CNNs are primarily used for pattern recognition with images [13] and they utilize their convolutional filters to extract the information from images [12]. These filters are matrices with specific values in their cells that are superimposed on the input image, which is itself a matrix (table) of values, with different values saved for each pixel (cell), in order to perform a series of calculations. These computations are useful for image manipulation (sharpening or blurring for example) or feature extraction: this second operation consists in the recognition of simple or aggregated geometrical features in the image (for example the human eye would be an aggregated feature composed of simpler geometrical features like circles, ellipses and edges).

In general, this kind of neural network works as follows: the first layer collects the inputs, the intermediate (or inner/hidden) layers perform the elaboration, and the last layer provides the output. CNNs are composed of three types of layers: the convolutional layers, the pooling layers and the *fully connected* layers [13]. The general representation of a CNN model can be seen in Figure 1. Close to the input, there are the convolutional layers: they perform the convolution of matrices (the "convolutional filters") against the image: these convolutional filters are moved across the image and applied to different areas of the image, and at each of these steps the convolutional operator (very similar to a "dot product" operator) is performed between the elements of that area of the image matrix and the values of the convolution filter (Figure 2). The values in the convolution matrix are adjusted during the learning phase by the algorithm. The output of the convolution operation is then defined as a feature map [14] (see Figure 3). Pooling layers will then simply perform down-sampling along the spatial dimensionality of the given input, further reducing the number of parameters [13] in order to guarantee computation in a feasible amount of time. After the series of convolutional and pooling layers, the resulting features' maps undergo flattening, and these matrices are converted into one column vector (in the flatten layer). The last section of a CNN (the set of fully connected layers) usually performs the classification task.

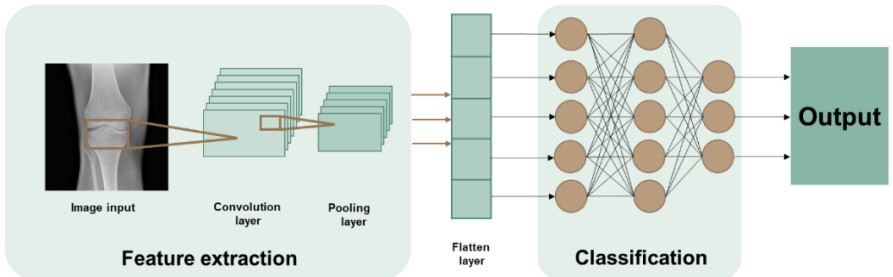

**Figure 1.** Graphical representation of a CNN. The first step is the feature extraction that happens thanks to the convolution filters; then, after flattening, the classification task is done by the fully connected layers.

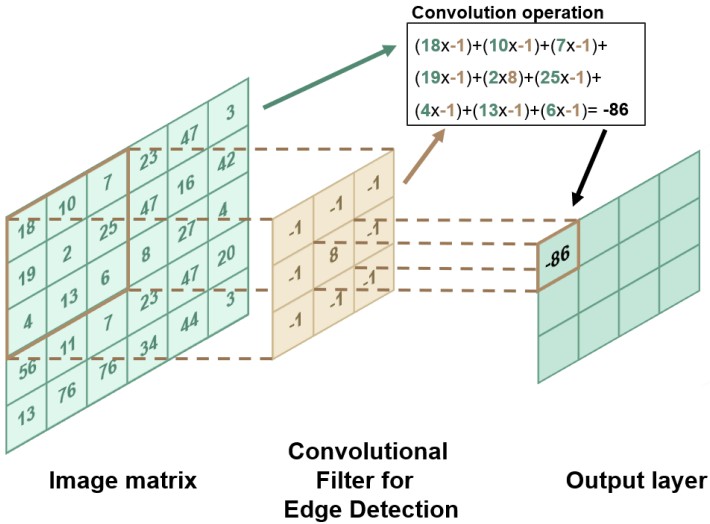

**Figure 2.** Graphical representation of a CNN. The first step is the feature extraction (edge detection in this case) that happens thanks to the convolution filters; then, after flattening, the classification task is done by the fully connected layers. The numerical values in the matrix on the right are the grey scale values of the image.

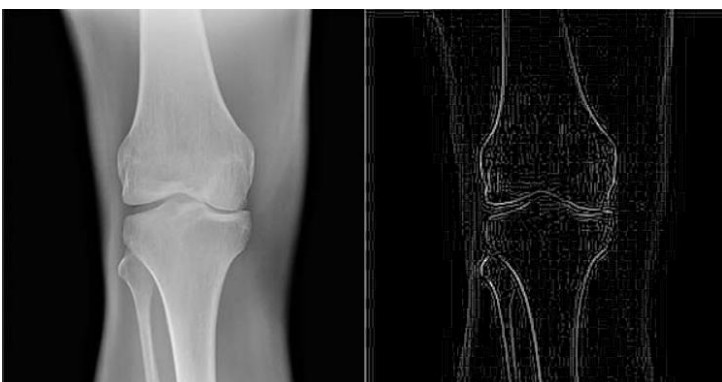

**Figure 3.** The image on the right side is the result obtained by applying an edge detecting convolution filter on the image on the left side. The structure of the filter is showed in Figure 2.

Often, to increase the overall efficiency [15], the number of hidden layers in the CNN is increased by a large amount. The result is a more complex neural network called **Deep Convolution Neural Network (DCNN)**. Another variation of a CNN is the **EfficientNet** model where the image resolution, width and color range (or depth) are uniformly increased as you proceed in the layers from input to output. This differs from the classic CNNs where these increases are arbitrary.

The level of complexity of a CNN model is dependent upon the size and the characteristics of its training dataset; it usually requires a large training data set because the more the investigated pattern or study aim is complex, the more extensive the input data have to be [10]. This large amount is of paramount importance in order to avoid overfit issues: Overfitting happens when the model is too closely aligned to the limited set of training datasets, and as a result the model becomes be useful in reference only to its initial data set and not to any other data sets, not achieving the aim of recognizing the patterns beneath the data [13].

Complex models tend to be more prone to overfitting when trained with small datasets, since they operate with many different parameters [5]; simpler models, on the other hand, are characterized by a smaller number of parameters and are thus limited in their ability to see non-existent patterns and relationships.

The increase in the number of training examples is usually done by means of data augmentation techniques, which consist in modifying the images' properties (such as saturation, contrast or brightness or geometric characteristics such as scale or orientation) in order to obtain different "altered" versions of the same image. Unfortunately, CNNs are generally bad at handling rotation and scaling, as they work with rectangular portions of the image with given dimensions.

Performance of CNNs is evaluated using data not present in the initial dataset used for training, which are therefore referred to as *test set* or *internal validation.* The core characteristics of these data, however, have to be identical to the ones used for model training and development (i.e., in terms of the same hospital and time period): indeed, algorithms generally perform poorly when *external validation* is performed (namely with datasets from different institutions [1]); therefore, the first validation is usually performed as an internal one. It is thus of note that a model's ability to generalize cannot be assessed only from internal validation: for example, in the orthopedic field, it is to be considered that different institutions could use a different labelling system for their radiographs, a different radiation dosage or different machines, and even the same one institution could undergo changes of these characteristics over time, making a previously internally validated ML model (trained on a certain type of images characteristic) not reliable anymore.

The **XGBoost** algorithm (Extreme Gradient Boosting) is a modern implementation of gradient boosting decision trees [10]. Boosting algorithms create weak learners (models), i.e., learners slightly better than random, and combine them into a strong learner in an iterative way [16,17]. These weak learners' models are defined "decision trees", which are thus described as "a supervised machine learning algorithm used for predictive modeling of a dependent variable (target) based on the input of several independent variables. They have a tree-like structure with the root at the top" [17].

## 3. Machine Learning Performance Indexes

When addressing a simple two classes classification problem, usually one class is defined as the "positive" one while the other as "negative". Without any implications on benefits or values, the positive class is usually the object of the study [5]. This distinction starts from the training samples given to the model, which can be either positive or negative; for example, in identifying a fracture, a positive sample could be a bone fracture vs. the negative sample being that of a healthy bone. Hence, once positive and negative examples are fed to the model, predictions made via the latter can be categorized and allocated as one of the following: true positive (TP), that is, a prediction of a positive result for a positive sample; true negative (TN), that is, a prediction of a positive result for a negative sample; false positive (FP), that is, a prediction of a positive result that in reality is negative; and false negative (FN), that is, a prediction of a negative result that is actually labeled as positive.

The formulas of the main performance indicators rely on these concepts. To better understand the performance of the models that will be discussed later, the main performance metrics must thus be introduced since only one index might not be enough to

describe the performance of a model. It is to be highlighted that the formulas reported herein are valid for a binary classifier, but the concepts can be extended to multi-class classification problems.

**Accuracy**: expresses how good a classifier is in its job of finding all the correct predictions related to the total number of predictions, i.e., the proportion of the total number of test samples in which the model identifies the positive labels [18].

$$Accuracy = \frac{TP + TN}{TP + FP + FN + TN} \tag{1}$$

**Sensitivity**: indicates how well the model is performing in terms of classifying the positive results related to the total number of predictions made regarding positive samples. It is also known as *true positive rate* [5] since it measures the ability of the model to predict positive results (all the actually pathologic subjects that were classified as pathologic by the model).

$$Sensitivity = \frac{TP}{TP + FN} \tag{2}$$

**Specificity**: quantifies how well the model is performing in terms of classifying the negative results related to the total number of predictions regarding truly negative results. It is also called the *true negative rate* as it quantifies how many true negatives (for example people that did not have a certain pathology that were actually recognized by the model as not having that pathology) were obtained.

$$Specificity = \frac{TN}{TN + FP} \tag{3}$$

**Precision**: computes what proportion of the positive predictions were actually true and is used when the goal is to limit the number of false positives [5]:

$$Precision = \frac{TP}{TP + FP} \tag{4}$$

**F1-score**: denotes the harmonic mean of precision and sensitivity [18]. It is used to assess whether our model has a high precision, a high sensitivity or both. In fact, the F1-score is high if both are high and thus optimized, it is medium if only one of the two metrics is optimized, and if it is low both metrics are low.

$$F1\ score = 2 \times \frac{Precision \times Sensitivity}{Precision + Sensitivity} \tag{5}$$

**ROC curve**: known as Receiver Operator Characteristic curve, it is plotted (as seen in Figure 4) on a graph with the "True Positive Rate" or "Sensitivity" on the *y*-axis and "False Positive Rate" or "1−Specificity" on the *x*-axis. The different values of the two rates are measured for different threshold levels in order to see how different values would influence the performance of the model. This curve also shows the trade-off between sensitivity and specificity.

**AUC**: is the Area Under the Curve between the ROC curve and the bisector. It measures the ability of a classifier to distinguish between the two or more classes, and it summarizes the ROC curve. AUC can vary in a range between 0 and 1, where 0 means that all positives have been evaluated as negatives and vice versa, while 1 means that all inputs were perfectly classified. Hence, classifiers with a curve closer to the top-left corner have better performance than those with a curve close to the 45° diagonal, since the model produces a high sensitivity while keeping a low false positive rate [5].

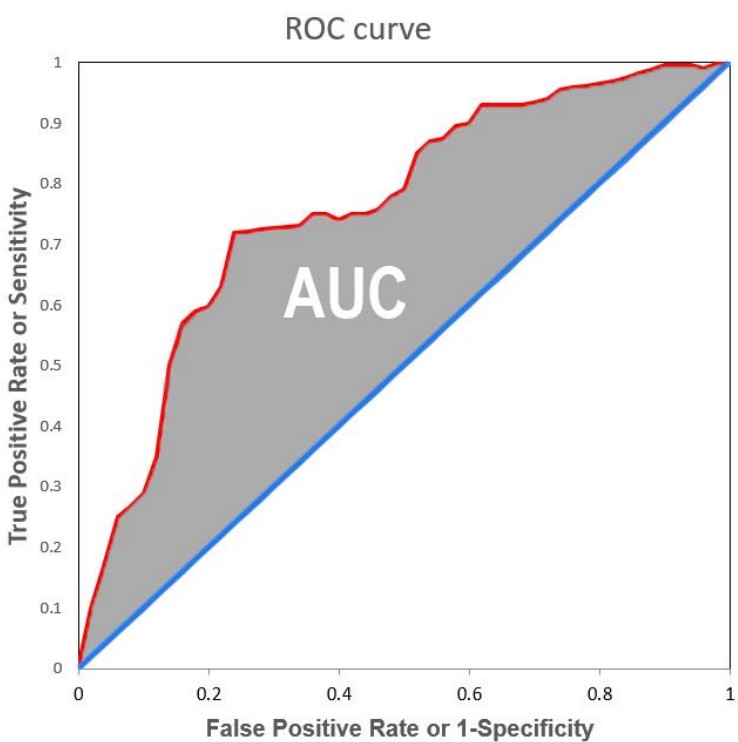

**Figure 4.** ROC curve in red and AUC in grey.

While accuracy is certainly the first reported performance index, it is not the only metric one should rely on in the case of an *imbalanced data set* (which is often the case in the orthopedic field). An imbalanced data set is one which presents much more examples of one of the classes than the other. This is where sensitivity and specificity are important since they specifically relate to the performance in classifying correctly one of the two classes. Sensitivity is more important when the request is to optimize the number of positive examples that are correctly classified, while the specificity is more relevant when the objective is to minimize the number of incorrectly classified negative examples. One thing to keep in mind is that sensitivity and specificity are often inversely related; thus, in trying to increase one, the other will decrease [5].

## 4. Diagnosing with AI

Over the past decade, the development of AI applications in orthopedics has focused primarily on diagnostics, mostly image interpretation [3]. A crucial function application for such an operation is computer vision and for this task CNNs are usually employed. See Table 1 for a list of the studies that were analyzed.

### 4.1. Hip Related Diagnosis

Cheng et al. [8] trained a diffusion convolution neural network (DCNN) to recognize hip fractures on frontal pelvic radiographs. The model underwent a pre-training on 25,505 limb radiographs. This operation is done to quicken the recognition ability of the final model since training on the required task does not start anymore with the model having randomly chosen weights, but ones that were already tuned to perform a similar task. The model then was trained on a set of 2804 frontal pelvic radiographs to detect hip fractures and obtained as result an accuracy of 91%, a sensitivity of 98%, a specificity of 84% and F1-score of 0.916. The model had only to confirm if there was a fracture on the radiograph without giving further information and, since it was trained to discriminate between healthy bones and fractures in the background of the bony architecture on radiographs, it might be unable to identify other lesions relevant for routine diagnoses [8]. Moreover, some bias is expected since differences in age, gender and injury severity score between

fractured and non-fractured patients were observed [8]. The performances were slightly lower than those obtained by radiologists and orthopedics, but this outcome nevertheless represented a promising result.

On the other hand, the CNN trained by Park et al. [18] outperformed the specialized doctors. Park and his team developed a neural network capable of recognizing bone tumors in proximal femur. Data augmentation was performed together with data pre-processing, so all text was removed from the radiographs and each radiograph was divided in left and right side by the longitudinal axis; the right side was then inverted, thus obtaining a total amount of 538 femoral images aligned with the same left side to help the machine by giving it a more homogeneous femur shape. Out of the multiple trained models, three variations of the EfficientNet model were chosen: EfficientNet-b1, b2 and b3. The EfficientNet-b2 model outperformed the other models and the human doctors with an accuracy that was higher than the one achieved by each of the four doctors involved in the comparison (two general orthopedic surgeons and two musculoskeletal tumor specialists [18]); moreover, the model sensitivity, precision and F1 were also superior (except for only one doctor). Image pre-processing was crucial for obtaining these results because of the standardized hip radiographs, but also the model that outperformed the other ones did it because of its smaller number of weights; this feature represents an advantage, when learning from small data sets, because a simpler model is limited in its ability to see non-existent patterns and relationships.

*4.2. Knee Related Diagnosis*

Liu et al. [19] constructed a CNN that had to recognize Tibial Plateau Fracture (TPF) from X-ray images and compared its performances with respect to expert orthopedic physicians, using as indicators accuracy and time spent on analysis. The algorithm was trained on a set of 916 JPEG files (training data set) and then tested on a set of 84 JPEG files (test data set) for algorithm validation. During the test, the algorithm had to label with a rectangle suspected fracture areas. The results were promising: the difference in accuracy of the machine with respect to the human radiologists (0.91 vs. $0.92 \pm 0.03$) was not statistically significant, while the time spent on average on each sample by the machine was 16 times faster than the one required by humans (0.55 s vs. $8.44 \pm 3.26$ s), and, more importantly, it is to be noted that the physicians were tested in a stress-free environment, which is rarely the case in a real life emergency department. This factor was indeed one of the reasons leading the authors of such a study to believe that the performance of their AI algorithm would be even better in the real clinical environment [19]. However, despite the excellent results, the trained algorithm could only recognize the fracture line without classifying it; this function, as the authors stated, will be added to the algorithm in the future. The other limitations found in this study were related to the fact that no normal knee radiographs were present in the data set, thus precluding the possibility of computing sensitivity and specificity. Additionally, the usage of only anteroposterior films may not be enough for a correct diagnosis of a TPF.

Another important injury related to the TPF that has become extremely common is the one involving the menisci, which are extremely vulnerable to injury due to their position and function in the knee joint [20]. The diagnosis of this type of injury is more complicated due to the presence of both bone structure and soft tissue. One of the many methods used for diagnosing is the magnetic resonance imaging (MRI), which has the advantage of being noninvasive and having high resolution for the soft tissues; the problem is, however, that the image resulting from the test would often appear blurred because of micro-movements of the limbs due to the patient, the breathing and/or the heartbeat [20]. In order to avoid these complications, Xie et al. [20] thought about using a machine learning approach for MRI image improvement: by using, in the process of MRI image reconstruction, a CNN-based algorithm, they removed "shadowy" parts and reduced blurriness in the under sampled images, giving then as output fully sampled cleaner MRI images. These images were then used for diagnosing and were later confirmed during the surgery itself. The

results showed an accuracy of 95.3% and a sensitivity of 96.9%, meaning that this CNN-based MRI optimizing algorithm could be considered effective for the diagnosis of tibial plateau fracture with meniscus injury, but the sample size used has to be considered, as the data set is not big enough to strengthen the results of the study.

A different application of AI was developed by Ghose et al. [21], who trained a CNN to recognize different total knee implant models, information that can become vital during preoperative planning before partial revision surgery (where only one or two components of the implant have to be changed). In this case the task to perform is different, and quite fitting for AI: while humans with good experience can generalize quite well characteristics of different objects/situations/conditions from different fields of knowledge and can recognize them with a high accuracy, they cannot process large amount of data like machines do. That is the case of the data of the implant's producer, model and general characteristics. If the surgeon is not familiar with the manufacturer and model of the implant, the routine procedure is to email the radiographs of the patient to medical representatives of the manufacturing companies for identification [21]; therefore, this identification can become an expensive and time-consuming operation. To tackle this problem, 878 images of six different models coming from five different manufacturers were used to train and validate a CNN that could perform the recognition task. Data augmentation was used [21] to increase the number of samples and a Contrast Limited Adaptive Histogram Equalization was used to increase the contrast of the images. The model performed the task with an accuracy of 96.66% despite the lack of a large number of example radiographs for each implant type. Additionally, the available data set samples were taken from only two different institutions and from different textbooks, thus no real external validation was done due to the difficulty of obtaining external clinical data.

*4.3. Other Orthopaedic Related Diagnosis*

Blüthgen et al. [11] developed a deep learning system aimed to diagnosticate distal radial fracture (DRF) using anteroposterior and lateral view radiographs. To build a model, an image analysis software (ViDi Suite Version 2.0) was used. After testing different models with various hyper-parameter combinations, two models (referred to as M1 and M2 from now on) were selected, as they gave the highest AUC values. Both the resulting models had to assign a value ranging from 1 to 0 where the first extreme meant that a defect area was found while 0 meant that nothing suspicious was noticed on the image by the machine. This score was coupled with a heat map on the image indicating the ROI, where the red zone was the one where the machine thought the probability of finding the fracture would be the highest. The resulting area was, as usual, compared to the ROI outlined by two expert radiologists and two test data sets were used, an internal data set with radiographs coming from the institution and an external data set comprising radiographs coming from different institutions. The deep learning models had a good tolerance against non-pathologic alterations of the radius and were capable of detecting fractures of DRF with near-human performance [11] with M1 performing with an AUC of 93% when using anteroposterior views, 94% when using the lateral views and an AUC of 95% when a combined version of the two views was used to train the machine, while M2 performed similarly with an AUC of 95%, 94% and 96% respectively. The AUC and also sensitivity and specificity were slightly lower for an external test data set, meaning that the model had more problems diagnosing DRF on radiographs coming from other institutions. Radiologists performed slightly better on the external data set but that could be also caused by a selection bias since external images were selected whether they were high quality or not.

Another diagnosing model for distal radial fractures was developed by Gan et al. [22]. Firstly, a Region Based CNN was trained to identify the distal radiuses on anteroposterior wrist radiographs as the region of interest (ROI) and then validated by two expert radiologists. Then the cropped and augmented images were sent to the Inception-V4 model that trained on a data set of 2040 images and tested on a set of 300 images equally divided

between radiographs with presence of DRFs and absence of DRFs. The results were an AUC of 0.96, an accuracy of 93% and a sensitivity of 90% a similar diagnostic capability to that of the tested orthopedists and superior performance to that of the tested radiologists. Many of the mistakes were committed on anteroposterior radiographs that displayed an absence of apparent fracture traits, features that were instead clearly visible from lateral radiographs corresponding to the anteroposterior images [22]. That is why the authors claim that the sensitivity will increase when the model will be given both types of radiographs.

## 5. Prediction with AI

The other important aspect of AI, and one of the most common types of tasks for a ML model, consists in its prediction functionalities. In orthopedics for example, regarding value metrics, ML methods have been used to predict the length of hospital stay, hospitalization charges and discharge disposition [23]: all aspects that may translate into improved patient care, reduced surgeon burnout and controlled resource costs if properly addressed [24]. It is then relevant to mention that this approach can also be used to address musculoskeletal surgeries, such as the identification of surgical candidates for a knee arthroplasty and their direction of referrals to the appropriate orthopedic surgeon [25].

### 5.1. Surgery Prediction

This problem has indeed been addressed by Houserman et al. [25] with the use of an EfficientNet-b4 that was used to identify if a patient was a candidate for surgery and if the type of surgery was either a total knee arthroplasty (TKA) or a medial unicompartmental knee arthroplasty (UKA).

The total data set consisted, for each of the 2767 patients, in three radiographs (before data augmentation): the weight-bearing antero-posterior, the lateral and the patellar views. Relying only on these three views, commonly ordered by primary care providers, the machine performed the task with an 87.8% accuracy. Anteromedial osteoarthritis is the main primary indication for UKA [25], and thus an even more significant outcome is that the result was obtained without the need of first identifying if the patient suffered from this pathology (hence without relying on a valgus stress view, which would have required a well-versed staff in order to be obtained). One limitation of the study, however, is that the information on which the model was trained was deeply based on the decision of the surgeons who performed the operations in the data-set, which are therefore taken as the "ideal choice"; it is, however, important to highlight that the follow-up data of the patients used to populate the training set indicated extremely few post-operative complications, therefore the choices made were considered as acceptable to represent the ideal result.

Another TKA related problem is the access disparity, caused by the fact that knee surgeons caring for patients who would benefit from TKA are incentivized to "cherry pick" healthier patients and "lemon drop" those with increased comorbidities and case complexity that would result in higher costs. That is why Ramkumar et al. [26] trained an artificial neural network to analyze 15 major attributes related to the patient in order to predict the length of stay, charges and costs, and patient disposition. The 15 attributes were age, gender, ethnicity, race, type of admission, whether the admission was from the emergency department, All Patient Refined (ARP) risk of mortality, APR severity of illness, number of associated chronic conditions and diagnoses, comorbidity status, whether the admission was on a weekend, hospital type, income quartile of the patient and whether the patient was transferred from an outside hospital. The outputs, instead, were divided as usual in only two per type. For the length of stay and charges and costs, the model had to specify if the predicted value was higher or lower than the 50th percentile (to enhance generalizability) and, for the patient disposition, if the patient would have been treated at home after the operation or not. After that, a risk-based patient-specific payment model was developed assessing the financial probability of over-cost related to APR risk of mortality. The APR score is part of the All Patient Refined-Diagnosis Related Group (APR-DRG)

methodology that was developed by 3M to allow analysis of outcomes across large cohorts for a given diagnostic group [27].

This study actually included external validation, since training data were taken from the National Inpatient Sample database with information coming from multiple hospitals while the test data came from the Orthopedic Minimal Data Set Episode of Care (OME), which contained information from 11 different hospitals. Due to the lack of inpatient charges in the second database, external validation was possible only for the first two outputs. After the training, length of stay prediction had an internal accuracy of 75.3% and AUC of 74.8%; after external validation, instead, these values improved with an accuracy of 80% and AUC 83.2%. Similar results were obtained for the inpatient discharges while discharge disposition achieved the lowest results, meaning that other patient-level variables needed to be included. Even though all performance indexes are inferior to 90%, the model was not only generalizable but also more reliable and responsive giving higher accuracy and AUC after being applied to the external OME data set [26]. At the same time the risk-based patient-specific payment model predicted an approximately 83% increase in the surgery cost for patients with extreme APR risk of mortality and a 22% increase for major risk patients. The authors proposed different future studies that could improve the model to produce more accurate predictions of duration of time and costs, since knowledge of these variables facilitates preoperative alignment between patient, surgeon, hospital and payer to allow for fair arbitration before knee surgery [26].

*5.2. Prediction of Post-Operative Complications*

Among others, the problem of complications following a TKA (like many other types of arthroplasties) also has a high impact on costs, since revision TKA is much more demanding and associated with higher costs and inferior outcomes. Thus, an early identification of patients at risk for revision is a factor becoming increasingly relevant [10]. The problem of predicting post-operative complications and of irregular durations of the surgery has been tackled by Hinterwimmer et al. [10]. An XGBoost algorithm was chosen to be trained on a data set of 864 patients with only 54 cases of complications and 99 cases of irregular duration of surgery. To tackle this class imbalance problem, a loss weighting was applied by imposing a higher loss weight to the class with fewer samples. The problem was then transformed into a binary classification problem, trying to identify whether a patient had at least one complication or none. The results for complication prediction were an accuracy of 92%, sensitivity of 34.8% and a specificity of 95.8%; the prediction of irregular surgery duration, instead, returned an accuracy of 93.4%, a sensitivity of 74% and a specificity of 96.3%. It is evident how having a slightly less imbalanced data set for this second problem (11.5% of the cases vs. 6.3% of the cases) beneficially impacted sensitivity. In this application, a "feature importance" was also computed, giving a ranking of the main features that were used by the model to classify the inputs, based on how useful they were at predicting a target [10], i.e., providing an indication on the extent to which a variable had been weighted in the ML model. However, it does not implicate causality nor unbiased associations [10]. The two major takeaways of the paper were that collaboration between data scientists and surgeons is paramount for the clinical interpretation of results and that the inclusion of post-operative data might be useful in predicting more complex outcomes such as early revisions.

Problems like the one presented previously can obviously also be encountered after the total hip arthroplasty (THA), where dislocation is the most common early complication and one of the main indications for revision surgery [9]. To assess the hip dislocation risk, Rouzrokh et al. [9] trained and tested a CNN on a data set of 97,934 antero-posterior pelvis radiographs taken at least 1 day after the surgery and at least 1 day before the possible dislocation. The performance was assessed using a 10-fold cross-validation, which means that the total data set was split into 10 groups and for each iteration one group was used as a test data set while all the others were used as training data sets. The model performance then was summarized using the scores each model received. The result with YOLO-V3

(which is a CNN) was an overall mean average precision of 99.2% between the right and left pelvis: this means that approximately 99% of the patients that had a dislocation in the future were identified by the model. One factor to keep in mind is how the model possibly relied on different imaging features when applied to different genders; this element, together with the different number of male and female dislocations in the data set, led to the different performances found between the two genders. All the choices were motivated by a saliency map and the future usage of other relevant factors and other X-ray views was suggested.

Table 1 summarizes the studies that were analyzed and their field of application.

**Table 1.** Research studies analyzed.

| Reference | Title | Application |
|---|---|---|
| Cheng et al., 2019 [8] | Deep learning for Detection of Complete Anterior Cruciate Ligament Tear | Diagnosis |
| Park et al., 2022 [18] | Artificial intelligence-based classification of bone tumors in the proximal femur on plain radiographs: System development and validation | Diagnosis |
| Liu et al., 2021 [19] | Artificial Intelligence to Diagnose Tibial Plateau Fractures: An Intelligent Assistant for Orthopedic Physicians | Diagnosis |
| Xie et al., 2021 [20] | Deep Learning-Based MRI in Diagnosis of Fracture of Tibial Plateau Combined with Meniscus Injury | Diagnosis |
| Ghose et al., 2020 [21] | Artificial Intelligence based identification of Total Knee Arthroplasty Implants | Diagnosis |
| Blüthgen et al., 2020 [11] | Detection and localization of distal radius fractures: Deep Learning system versus radiologists | Diagnosis |
| Gan et al., 2019 [22] | Artificial intelligence detection of distal radius fractures: a comparison between the convolutional neural network and professional assessments | Diagnosis |
| Houserman et al., 2022 [25] | The Viability of an Artificial Intelligence/Machine Learning Prediction Model to Determine Candidates for Knee Arthroplasty | Prediction |
| Hinterwimmer et al., 2022 [10] | Prediction of complications and surgery duration in primary TKA with high accuracy using machine learning with arthroplasty-specific data | Prediction |
| Ramkumar et al., 2019 [26] | Deep Learning Preoperatively Predicts Value Metrics for Primary Total Knee Arthroplasty: Development and Validation of an Artificial Neural Network | Prediction |
| Rouzrokh et al., 2020 [9] | Deep Learning Artificial Intelligence Model for Assessment of Hip Dislocation Risk Following Primary Total Hip Arthroplasty From Postoperative Radiographs | Prediction |

## 6. Limitations of AI and Implications for the Future

Even if the results accomplished seem promising, showing their potential to make the work of the clinicians much easier, the limitations of such technology have always to be kept in consideration (together with its legal and ethical consequences) in order to correctly evaluate the possibilities and to properly integrate them in the health system. What follows is thus a deeper look into these limitations and their relative consequences.

### 6.1. External Validation and the Change in the Clinician's Working Routine

Machine learning is nowadays in its early stages, and standardized approaches are not yet established. This is particularly true for the external validation of the model [1,10]. The results obtained with the various models seem to be positive, but most of them relied only on data coming from one or maximum two institutions, while external validation or the measure of the model's ability to generalize is a crucial point for its broader implementation.

The model also requires proper supervision and data capturing during the treatment [28]. For the general validation, a close collaboration between a surgeon and a data scientist must therefore occur, in order to not only correctly evaluate the validity of the output, but also to keep the ML model tuned and well maintained in time [3,10]. Unfortunately, all these operations are still expensive, posing a big obstacle to a faster implementation of AI in the medical field.

### 6.2. Data Limitations and How to Collect Data

The first problem resides thus in the fact that an ML model strictly depends upon its data set. This means that a biased data set will certainly affect the performance of the model in a negative way [23]. Moreover, a machine learning model cannot generalize a concept or a structure, hence it finds it more difficult to detect an anomaly that is too rare in the set of examples of an imbalanced data set [11]. A possible solution could be to include a much bigger and balanced data set, with information structured in a way that is most suitable for the ML model in order to also avoid overfitting (as already discussed previously). This also means that the training data should be representative of the general population independently from gender, race, and cultural and economic backgrounds. The most practical and efficient way to obtain such results (and maybe the only one) is the concept of *data sharing*, i.e., a collaboration between institutions to create pooled data sets with information coming from different realities [9]. Data labeling also has to be monitored and standardized since it is an operation performed by a human that inevitably introduces subjectivity and bias. Finally, data privacy and protection are of utmost importance going forward [29]; therefore, all these data-managing operations have to be done in an ethical manner by assuring that all privacy norms are respected in the process and patients' consent is given. Unfortunately, this process takes extensive periods of time and limits the volume of data that can be gathered by the various registries [29].

### 6.3. Black-Box and Responsibility Issues

The second problem is related to the model being a "Black-Box", meaning that no real explanation and motivation of the result can be directly given by the algorithm. Moreover, if any miscalculation occurs, the medical responsibility cannot be identified since relevant laws are missing [19]; furthermore, if the medical figure cannot explain to the patient the reasons behind the results given by the model, the patient's trust in the medical figure would erode, making it also difficult to have informed consent [30]. The responsibility issue is also very important because technologists are not obligated by law to be accountable for their actions [31]. That is why there is still a debate on whether the technicians should be held responsible for the results given by the machine or not. What some suggest is that it is the underlying data given to the algorithm that should be held responsible [31]. Therefore, there is the need for clear and trustworthy outputs that can be seen in the effort to create *explainable artificial intelligence* (XAI), that could justify its decisions to a human collaborator. *Explainability* must not, however, be confused with *interpretability*: the former is the ability of the machine to describe the processes that brought the results, while the latter is the ability to provide understandable reasons behind individual decisions that clinicians can use to make judgements [32]. Some techniques used to increase interpretability were adopted in the cited papers [9,19–21,33], including features such as importance, region of interest, saliency map, and heat map: the first one assigns a score to input parameters, based on how useful they are at predicting a target; the second underlines the variables that are considered the most relevant for the task by the machine, and the third and fourth ones instead underline, in different ways, the key features of an image for the classification task. In this way, it is then possible to obtain a motivation for the result without actually knowing the processes that brought it about.

*6.4. Automation and the Patient–Physician Relationship*

Automating diagnoses, recommending appropriate treatment approaches and predicting the outcome following a treatment all fall within the scope of AI [34]; in many of the studies it is assured that the final goal is not to replace the clinicians, but the relative concern is real and understandable.

As seen, the results are promising but they regard relatively simple tasks and anything different from that specific duty cannot be achieved by that AI model. Additionally, AI lacks the creative thinking required in all of the medical subfields [28] where making good clinical judgements cannot take the form of an algorithm, since statistical methods cannot grasp the complexities of human cognition and behavior [35]. That is why many impartial observers consider AI applications only as a way to assist clinical decision-making, not yet being in position to replace it [36]. Many others, instead, have a different type of concern more relevant to the present day, regarding the fact that AI could erode the relationship between patient and physician. Since AI cannot understand emotions and human thought [28], there is the risk of dehumanizing medicine [35]. The more diagnosis and decision-making processes rely on such a technology, the more this fear of dehumanization becomes true as this approach reduces the human interaction between physician and patient. This could thus bring not only to a deterioration of skills and lack of pleasure in the job on the physician's side, but also a reduced questioning of the results given by the model (from both patient and physician's side) due to the high accuracy of the machine [31]. Medicine cannot be reduced to mere scientific knowledge and implementation of technology, but must also include the humanistic considerations that are intrinsic to good care [35].

## 7. Discussion

This paper aimed to analyze the latest achievements of Artificial Intelligence in the field of orthopedics, providing the reader with an overview of all the possibilities of this technology. As shown, machine learning models can indeed provide meaningful assistance to surgeons in their decisions or even help in diagnosing a certain pathology.

For the purpose of diagnosing, different studies were analyzed in which models were trained to recognize fractures in various parts of the body such as the hip and the knee. Other models, instead, were trained to improve time-consuming tasks such as MRI image improvement or implant model and characteristics recognition. The models were able to perform the tasks similarly if not superiorly to their human counterpart, according to performance indexes as accuracy, sensitivity, specificity, precision, F1-score and AUC. Admirable results were thus achieved, but it must be kept in mind that most of the time the models had to deal only with a simple binary recognition (as an example, "fracture is present" versus "no fracture found") and therefore they were not able to perform specific and exhaustive diagnosis.

Regarding prediction, instead, in the cited studies various ML models were trained to predict different aspects related to surgery such as the following: length of stay after TKA surgery, evaluation of suitability of a candidate for TKA or UKA based on three different views of the knee joint, surgery cost increase for patients with comorbidities with respect to patients with only the primary condition, and post-operative complications after total knee or hip arthroplasties. These tasks were performed satisfyingly as well, and some models were even externally validated (thus assessing the generalizability of their algorithm).

It is to keep in consideration that all the models addressed are strictly dependent on the data fed to them, which should therefore be of the right quantity and also of the right quality. The data fed to the algorithm should be unbiased, hence gathered and selected by competent figures able to ensure a proper representativeness of the general population. No discrimination should be permitted. Since the machine can amplify the intrinsic biases of the data, moreover, the data selected to train the model should be considered responsible for the results given by the machine. This theme of responsibility in medical decision-making is crucial and for this reason is the main topic of current scientific discussions:

who would be responsible for the mistake in the decision-making process once man and machine cooperate is indeed still not clear, and inherent laws are missing.

All of the models were tested and evaluated on their performance, but their integration in the medical environment was not assessed: this means also that the interaction between clinician and machine was not analyzed, together with the impact of such a technology on day-to-day work. What was instead considered in some of the addressed studies is that a strict collaboration between the clinician and a data scientist is fundamental, since the medical figure alone would need knowledge and skills in a field different from their own in order to maintain the machine and validate the results given by the model.

It is still finally to be kept in consideration the fear that, while these models are still far from substituting the human clinician, they could already interfere in the patient–clinician interaction, which is crucial for the correct evaluation of the patient's situation and for the decision-making process. Moreover, the clinicians could over time rely less and less on their competence and skills as they become more complacent about the results given by the machine, instead of analyzing them with a critical approach.

### 8. Conclusions

To conclude, this article analyzed multiple studies related to applications of artificial intelligence in the field of orthopedics, underlining the peculiarities, the pros and the cons of each study and, more broadly, of the technology itself.

Even if the approach and the methods used nowadays have yet to be standardized and further optimized, the possibilities offered by this powerful tool are almost endless as they range from the assistance of clinicians to the prediction of outcomes of surgery: for this reason, the potentialities of the use of AI in orthopedics have to be kept in great consideration for the years to come, without, however, losing sight of their drawbacks in order to prevent unhappy consequences.

**Author Contributions:** Conceptualization, B.I. and E.B.; investigation, Y.R.; writing—original draft preparation, Y.R.; writing—review and editing, Y.R., E.B. and B.I.; supervision, B.I. All authors have read and agreed to the published version of the manuscript.

**Funding:** This research received no external funding.

**Institutional Review Board Statement:** Not applicable.

**Informed Consent Statement:** Not applicable.

**Data Availability Statement:** Not applicable.

**Conflicts of Interest:** The authors declare no conflict of interest.

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
