# Peer review of "The Use of Artificial Intelligence in Orthopedics: Applications and Limitations of Machine Learning in Diagnosis and Prediction"

_applsci, doi:10.3390/app122110775_

Round 1

Reviewer 1 Report

The paper presents the state of the art of the application of AI- ML to Orthopedics. It provides a useful critical analysis (6) and a fruitful discussion (7).

As regard the AI-ML methods introduction, in my opinion a review paper should provide a deeper general  introduction to AI – ML methods, without going  into the specific application (note that  Ref 4 seems to be the only general reference and it refers to CNN only), but considering the most suitable methods in orthopedics in their generality. Moreover some critical aspect depicted in par 6, can be introduced here, before going into application details.

For example some points that in my opinion are missing in the general part are:

-          General methods in ML not involving CNN: for example see lines 278: M1 and M2,, Hyperparameters; 388: XGBoost algorithm (an algorithm that oper-388 ates on decision trees, differently than CNN)

-          Specific details of CNN based methods with reference to the ones than will be presented in the various applications: for example see lines 175 DCNN or 200 Efficient Net bx

-          Input data and training issues: for example see lines 167: polarized data sets ;  210: complex vs simple models overfitting and other issues; 252: data set optimal dimension for validation; 391

-          Validation problems, gold standard (339) and other related aspects: for example line 190 radiologist and orthopedics 339,401: Since this result indicates to what extent a variable has been 400 weighted in the ML model, it can be important to assess the validity of the model. ,412,434,440

A synthesis of the analyzed state of the art could help the reader. A table or a scheme could be useful for presenting the literature analysis in an aggregate way giving at a glance peculiarities and drawbacks of the methods/applications.

Other points:

178: 'Unrelated' means that the validation in [3] is based on 'related' images? what does it mean? are the authors referring to correlation between test images? Perhaps some details regarding this problems in validation can be added (see point before)

180: why a pre-training on a different goal (part identification) can reduce the training sample size for a different goal (hip fractures)?

Figure 2: the effect of a specific filter can be added to demonstrate filter effect on the image.

Figure 3 can be smaller since the information content is rather low

Line 144 –‘1’

Author Response

The authors thank the reviewer for their comments; following, said comments are reported in bold and the relative response in italics.

The paper presents the state of the art of the application of AI- ML to Orthopedics. It provides a useful critical analysis (6) and a fruitful discussion (7). As regard the AI-ML methods introduction, in my opinion a review paper should provide a deeper general  introduction to AI – ML methods, without going  into the specific application (note that  Ref 4 seems to be the only general reference and it refers to CNN only), but considering the most suitable methods in orthopedics in their generality Moreover some critical aspect depicted in par 6, can be introduced here, before going into application details.

The relative introduction and machine learning sections were modified and improved adding further info and references.

For example some points that in my opinion are missing in the general part are:

-          General methods in ML not involving CNN: for example see lines 278: M1 and M2,, Hyperparameters; 388: XGBoost algorithm (an algorithm that oper-388 ates on decision trees, differently than CNN)

-          Specific details of CNN based methods with reference to the ones than will be presented in the various applications: for example see lines 175 DCNN or 200 Efficient Net bx

-          Input data and training issues: for example see lines 167: polarized data sets ;  210: complex vs simple models overfitting and other issues; 252: data set optimal dimension for validation; 391

-          Validation problems, gold standard (339) and other related aspects: for example line 190 radiologist and orthopedics 339,401: Since this result indicates to what extent a variable has been 400 weighted in the ML model, it can be important to assess the validity of the model. ,412,434,440

Further sentences and relative references were added according to the reviewer’s suggestions, addressing these methods and topics also in the initial sections.

A synthesis of the analyzed state of the art could help the reader. A table or a scheme could be useful for presenting the literature analysis in an aggregate way giving at a glance peculiarities and drawbacks of the methods/applications.

The authors appreciate the reviewer’s suggestion, a table (Table 1) with the most significant studies referenced, divided according to the application of the AI-ML techniques, was added to provide the reader a straightforward overview of the recentness and use of these approaches.

Other points:

178: 'Unrelated' means that the validation in [3] is based on 'related' images? what does it mean? are the authors referring to correlation between test images? Perhaps some details regarding this problems in validation can be added (see point before)

180: why a pre-training on a different goal (part identification) can reduce the training sample size for a different goal (hip fractures)?

The authors apologize for the lack of clarity of these sentences; a complete rephrasing was performed to address this issue.

Figure 2: the effect of a specific filter can be added to demonstrate filter effect on the image.

A further figure (now Figure 3) was added to show the effects of the filter, accordingly to the suggestion.

Figure 3 can be smaller since the information content is rather low

The figure was removed in agreement with reviewers’ suggestions.

Line 144 –‘1’

The ambiguity of this line was corrected by adding “” in order to clarify the meaning of the sentence.

Reviewer 2 Report

Authors submitted a paper to review the use of Artificial Intelligence in Orthopaedics: applications and limitations of Machine Learning in diagnosis and prediction. Although the study is a narrative review, it doesn't consider all relevant studies. The structure is also confusing as the readers cannot find a logical procedure for the discussed topics. Meta analysis and risk of bias also should be considered to have a good review paper as the authors didn't explain how the studies were selected. 

Author Response

The authors are saddened to see these results from the reviewer, and we hope that the revised version of the paper may change their overall 1-star evaluation; since also the English language use was reported to be non-sufficient (by this reviewer only), a further check from a native speaker was performed and the manuscript was revised accordingly. Following, comments are reported in bold and the relative response in italics.

Authors submitted a paper to review the use of Artificial Intelligence in Orthopaedics: applications and limitations of Machine Learning in diagnosis and prediction.

Although the study is a narrative review, it doesn't consider all relevant studies.

Further status of the art research was done and additional references were added according to the reviewers’ suggestions to improve the manuscript, increasing their number from 27 to 36.

Being a narrative review, however, the main aim of the article is to provide an overview of the different techniques available and their application in the field of orthopaedics, providing eventual studies from the literature as examples of said use. It is not the scope of the paper to perform a systematic review, therefore the approach used in the writing is consequently different. 

The structure is also confusing as the readers cannot find a logical procedure for the discussed topics.

The authors apologize for the lack of clarity, the manuscript was modified accordingly to the suggestions;

it is now clearly separated paragraphs addressing: the concepts and techniques, the performance evaluation indexes, the two main uses in orthopaedics, the limitations of the addressed techniques and finally the discussions paragraph who addresses and correlate all the previous ones.

Meta analysis and risk of bias also should be considered to have a good review paper as the authors didn't explain how the studies were selected.

The studies were selected with the aim of providing the reader with a series of examples to be paired with the different techniques addressed in this narrative review; it was therefore out of the scope of this paper to perform a meta-analysis (which aims to provide objective and statistical results), and therefore the approach used in the study selection and article writing was mainly aimed to offer the reader a broad overview to provide them the tools to subsequently go more in depth. 

Reviewer 3 Report

Dear Authors, the manuscript is intriguing and well structured, but I must suggest to enrich the number of bibliographic references and to shorten certain sections:

Half of the abstract is an introduction to the background. I suggest providing some preliminary data in the abstract, not just future perspectives and preambles for the manuscript.

33 references are missing

34-37 paragraph equal to the abstract, however the objective by convention must be moved to the end of the introduction

38-42 I suggest removing this revision design description

45-47 I suggest removing these declarations of intent

47 are not cited

I would actually suggest the description of machine learning approaches first: regression (https://doi.org/10.3389/fonc.2022.843611 ), cluster (https://doi.org/10.3390/jcm11123505) and classification (https://doi.org/10.1148/radiol.2021204289). Successivamente descriverei i modelli

57 completely missing references

“Is considered the gold standard for imaging approach”, but reference is necessary

100 Paragraph very captivating and easy to approach, I only ask you to justify any sentence with a bibliographic reference and to remove the confusional matrix because it is more complex than that, there would be a discussion on decision boundaries, I would recommend describing only the quality indices as already exposed.

The narrative review is built well. I would reduce the discussion and above all by convention the limitations go to the end of the manuscript, preceding the conclusions.

I would also reduce the limitation section with insightful suggestions on the issues associated with this type of approach, and I would also emphasize the concept of over-fitting.

Author Response

The authors thank the reviewer for their suggestions concerning references and readability improvements; following, the reviewer’s comments are reported in bold and the relative response in italics.

Dear Authors, the manuscript is intriguing and well structured, but I must suggest to enrich the number of bibliographic references and to shorten certain sections:

Half of the abstract is an introduction to the background. I suggest providing some preliminary data in the abstract, not just future perspectives and preambles for the manuscript.

The abstract was written in order to represent an overview of the article content, which is a narrative review; for this reason, and to abide to the length restrictions, it was decided to articulate it as a description of the contents in terms of different sections rather than a list of the techniques addressed, but still mentioning the two main applications in orthopedics.

33 references are missing

The relative further references were added.

34-37 paragraph equal to the abstract, however the objective by convention must be moved to the end of the introduction

The section was modified accordingly to the reviewer’s suggestions.

38-42 I suggest removing this revision design description

45-47 I suggest removing these declarations of intent

The sections were modified accordingly to the reviewer’s suggestions, eliminating the relative sentences.

47 are not cited

I would actually suggest the description of machine learning approaches first: regression (https://doi.org/10.3389/fonc.2022.843611 ), cluster (https://doi.org/10.3390/jcm11123505) and classification (https://doi.org/10.1148/radiol.2021204289). Successivamente descriverei i modelli

57 completely missing references

“Is considered the gold standard for imaging approach”, but reference is necessary

The suggested references were added to the manuscript in the relative sections, and further ones were added to address this lack.

100 Paragraph very captivating and easy to approach, I only ask you to justify any sentence with a bibliographic reference and to remove the confusional matrix because it is more complex than that, there would be a discussion on decision boundaries, I would recommend describing only the quality indices as already exposed.

A bibliographic reference was added accordingly, and the confusion matrix was removed as suggested; the text was thus adapted to explain the concepts of the possible predictions.

The narrative review is built well. I would reduce the discussion and above all by convention the limitations go to the end of the manuscript, preceding the conclusions.

Discussion paragraph was reduced accordingly to the suggestions;

Concerning the positioning of the limitations, the decision was made as they represent limitations not of the article per se, but rather of the technology itself; for this reason, indeed, these limitations cover an important role in the subsequent discussion section and therefore it was considered more clear for the reader to address them in the presented order.

I would also reduce the limitation section with insightful suggestions on the issues associated with this type of approach, and I would also emphasize the concept of over-fitting.

Limitations were shortened when possible, and the concept of overfitting was addressed more in detail accordingly to the suggestions.

Reviewer 4 Report

I commend the authors for their research entitled "The use of Artificial Intelligence in Orthopaedics: applications and limitations of Machine Learning in diagnosis and prediction". The content is attractive, the topic is contemporary, the review is thorough and systematic, the conclusions are sound, and the manuscript is easy to read. I would only suggest to add two Tables with all the discussed research articles of the main two applications - diagnosis and prediction - summarized. 

Author Response

The authors thank the reviewer for their kind words. Their comment was indeed useful to improve the quality of the manuscript, and “Table 1” was added to summarize the articles of the relative categories.

Round 2

Reviewer 1 Report

After this revision the paper is ready for publication.

Author Response

The authors wish to thank again the reviewer for their contributions to the manuscript, which helped to improve it up to its current version.

Reviewer 2 Report

I believe this study doesn't have a good structure, with the lack of sufficient critical appraisal. There is no specific method for its review, nor clear outcomes. 

Author Response

The authors are saddened to read that, despite the effort put into the manuscript revision, no changes in the reviewer's opinion were achieved.

We hope however that the generally positive feedbacks received by the rest of the reviewers may nonetheless represent a litmus test of the overall manuscript latest implementation.

Reviewer 3 Report

I can suggest the manuscript's suitability for publication

Author Response

The authors thank the reviewer for their kind words and for their suggestions for the previous version of the manuscript.